# Monomeric C-Reactive Protein in Atherosclerotic Cardiovascular Disease: Advances and Perspectives

**DOI:** 10.3390/ijms24032079

**Published:** 2023-01-20

**Authors:** Ivan Melnikov, Sergey Kozlov, Olga Saburova, Yuliya Avtaeva, Konstantin Guria, Zufar Gabbasov

**Affiliations:** 1National Medical Research Centre of Cardiology Named after Academician E.I. Chazov, Ministry of Health of the Russian Federation, 15A 3-rd Cherepkovskaya Street, 121552 Moscow, Russia; 2State Research Center of the Russian Federation, Institute of Biomedical Problems of Russian Academy of Sciences, 76A Khoroshevskoye Shosse, 123007 Moscow, Russia

**Keywords:** monomeric C-reactive protein, mCRP, C-reactive protein, biomarker, atherosclerosis, inflammation, thromboinflammation, residual inflammatory risk

## Abstract

This review aimed to trace the inflammatory pathway from the NLRP3 inflammasome to monomeric C-reactive protein (mCRP) in atherosclerotic cardiovascular disease. CRP is the final product of the interleukin (IL)-1β/IL-6/CRP axis. Its monomeric form can be produced at sites of local inflammation through the dissociation of pentameric CRP and, to some extent, local synthesis. mCRP has a distinct proinflammatory profile. In vitro and animal-model studies have suggested a role for mCRP in: platelet activation, adhesion, and aggregation; endothelial activation; leukocyte recruitment and polarization; foam-cell formation; and neovascularization. mCRP has been shown to deposit in atherosclerotic plaques and damaged tissues. In recent years, the first published papers have reported the development and application of mCRP assays. Principally, these studies demonstrated the feasibility of measuring mCRP levels. With recent advances in detection techniques and the introduction of first assays, mCRP-level measurement should become more accessible and widely used. To date, anti-inflammatory therapy in atherosclerosis has targeted the NLRP3 inflammasome and upstream links of the IL-1β/IL-6/CRP axis. Large clinical trials have provided sufficient evidence to support this strategy. However, few compounds target CRP. Studies on these agents are limited to animal models or small clinical trials.

## 1. Introduction

Atherosclerosis and its complications, primarily coronary artery disease (CAD) and ischemic stroke, remain the leading causes of mortality and disability worldwide. The incidence of atherosclerotic cardiovascular disease and major adverse cardiovascular events (MACE) increases with age. The probability of the clinical manifestation of atherosclerosis and the development of MACE is determined by the total plaque burden [1]. The total plaque burden is characterized by the concentration and duration of exposure to circulating atherogenic apolipoprotein B (apoB)-containing lipoproteins [1]. As plaque burden increases, the probability of atherosclerotic cardiovascular disease onset increases [1,2]. Therefore, reducing low-density lipoprotein cholesterol (LDL-C) level, the main fraction of apoB-containing lipoproteins in the blood, is the mainstay of atherosclerosis prevention. The reduction in LDL-C levels is achieved through diet, lifestyle modification, and pharmacological treatment with statin monotherapy, statin in combination with ezetimibe or proprotein convertase subtilisin-kexin type 9 (PCSK9) inhibitors [3].

Large randomized clinical trials demonstrated that aggressive lipid-lowering therapy with statins in combination with PCSK9 inhibitors produced a formidable reduction in LDL-C levels, approaching extremely low values in some patients. For example, a subgroup of 504 patients in the FOURIER trial achieved an LDL-C level of 0.18 mmol/L [4], whereas a subgroup of 3357 patients in the ODYSSEY OUTCOMES trial achieved an LDL-C level of less than 0.65 mmol/L [5]. Nevertheless, the MACE rate observed in these patients was substantial [6], which could be ascribed to the large total plaque burden that had already accumulated in these patients prior to treatment [1,2].

The cardiovascular risk that persists despite aggressive lipid-lowering therapy and correction of modifiable risk factors is called residual cardiovascular risk [7]. One of its main types is the residual inflammatory risk resulting from low-grade inflammation in atherosclerotic plaques [8]. It is determined by the level of the main inflammatory biomarker C-reactive protein (CRP), measured using a high-sensitivity assay (hsCRP), with a value of 2.0 mg/L or more [9]. The hsCRP assay measures the level of the pentameric form of CRP (pCRP), which is produced in the liver under the stimulation by interleukin (IL)-6 [10]. There exists another form of CRP, monomeric CRP (mCRP), which is formed at sites of local inflammation through the dissociation of pCRP and, to some extent, local synthesis. mCRP is essentially different from pCRP in its functions [10,11]. mCRP has long remained the subject of basic research; however, in recent years, reports measuring the levels of mCRP in the blood of patients and healthy individuals have been published.

This review aimed to trace the inflammatory pathway from the NLRP3 inflammasome to mCRP in atherosclerotic cardiovascular disease. The role of mCRP in other diseases, such as cancer, venous thromboembolism, and age-related and neurodegenerative diseases, has been discussed in various reports [12,13,14,15]. Hereafter, the abbreviations CRP, pCRP, hsCRP, and mCRP may be used. CRP is used when referring to studies that did not distinguish between forms of CRP. pCRP represents the pentameric liver-produced form of CRP. hsCRP is pCRP measured using a high-sensitivity assay. Finally, mCRP represents the monomeric dissociated form of CRP.

## 2. NLRP3 Inflammasome and Inflammatory IL-1β/IL-6/CRP Axis in Pathogenesis of Subclinical Vascular Inflammation

Cellular immune responses play a key role in all stages of atherosclerotic lesion development, from fatty streaks to plaque rupture [16]. A detailed discussion of cellular immunity in the pathogenesis of atherosclerosis has been provided elsewhere [17,18]. Four types of cells are involved in the pathogenesis of atherosclerotic lesions: smooth muscle cells (SMCs), T-lymphocytes, macrophages, and neutrophils. The cytoplasm of myeloid immune cells, including macrophages and neutrophils, and SMC contains pattern-recognition receptors (PRRs) [19,20]. These receptors recognize exogenous pathogen-associated molecular patterns (PAMPs) and endogenous damage-associated molecular patterns (DAMPs). DAMPs arise from signals produced by injured cells and tissues in the absence of an exogenous pathogen [20]. One type of PRR is the nucleotide-binding oligomerization domain-like receptor (NLR) [21]. The NLRP3 receptor in this group recognizes non-pathogenic DAMPs. NLRP3 activation leads to the formation of a molecular complex, the inflammasome, in the cytoplasm of myeloid immune cells. The NLRP3 inflammasome is a cytoplasmic protein complex that serves as a molecular platform for the activation of the cysteine protease caspase-1 [21]. Caspase-1 proteolyzes three proteins: pro-interleukin (pro-IL)-1β, pro-IL-18, and gasdermin D. Pro-IL-1β is converted into the active proinflammatory form of IL-1β and pro-IL-18 is converted into active proinflammatory IL-18. The activation of gasdermin D can induce pyroptosis, an inflammatory cell death, with the ensuing release of cytoplasmic content into surrounding tissues. This is particularly observed in the apoptosis of foam cells in atherosclerotic lesions [22,23].

The NLRP3 inflammasome formation has been observed in a number of diseases characterized by sterile inflammation. Uric-acid crystals can activate the NLRP3 inflammasome in gout [24]. Furthermore, the NLRP3 inflammasome plays an important role in the development of abdominal aortic aneurysm [25], myocardial damage in ischemia-reperfusion injury [26], kidney damage in diabetes, gout, and acute renal failure [27]. Active oxygen species and free fatty acids activate the NLRP3 inflammasome in obesity [28] and diabetes [29], contributing to the development of insulin resistance [30].

In 2010, Duewell et al. demonstrated for the first time the activation of the NLRP3 inflammasome by cholesterol crystals [31]. CD36-mediated uptake of oxidized LDL by macrophages results in intracellular cholesterol crystallization. This disrupts phagocytosis and leads to the accumulation of cholesterol crystals in macrophage lysosomes [32]. Subsequently, cholesterol crystals damage lysosome membranes and are released along with the lysosomal protease cathepsin B. They interact with PRR in the cytoplasm and induce the NLRP3 inflammasome formation [33]. Calcium-phosphate crystals accumulated in calcified atherosclerotic plaques can also activate the NLRP3 inflammasome [34]. Moreover, a recent study showed that CRP could activate the NLRP-3 inflammasome via the nuclear factor kappa B (NF-kB) pathway [35].

The NLRP3 inflammasome activation triggers a central cascade of inflammatory signaling represented by IL-1β, IL-6, and CRP [36]. IL-1β and IL-6 are mainly produced by myeloid cells [37,38], whereas CRP is by hepatocytes [10]. IL-1β stimulates the release of chemokines and the expression of cell-adhesion molecules by endotheliocytes, facilitating leukocyte recruitment to the site of inflammation. IL-1β stimulates the proliferation of SMC and the secretion of chemokines and collagenases by macrophages, thus contributing to the destabilization of atherosclerotic plaques [39].

IL-1β induces IL-6 synthesis. IL-6 has a wide range of proinflammatory and anti-inflammatory properties. For example, IL-6 stimulates the chemotaxis of neutrophils and macrophages to the site of inflammation [40]. It also stimulates chemokine release and expression of cell-adhesion molecules by endotheliocytes, facilitating leukocyte recruitment and platelet activation [41]. IL-6 produces a proatherogenic effect by stimulating SMC proliferation and modifying LDL uptake by macrophages and SMC [42]. Simultaneously, it stimulates an increase in LDL receptor expression, acting in an anti-atherogenic way [43]. IL-6 exhibits anti-inflammatory action by inhibiting IL-1 and tumor necrosis factor-α (TNF-α) synthesis and increasing IL-1 receptor antagonists and soluble p55 TNF-α receptor release [44]. Despite both proinflammatory and anti-inflammatory effects, higher levels of IL-1β and IL-6 are unequivocally associated with increased cardiovascular risk [45]. IL-6 stimulates CRP production in hepatocytes.

## 3. Assessment of Residual Inflammatory Risk

### 3.1. C-Reactive Protein as a Biomarker of Subclinical Vascular Inflammation

CRP is the final product of the central inflammatory cascade. Owing to its excellent reproducibility, it is considered the main biomarker of inflammation that reflects the activity of the IL-1β/IL-6/CRP axis [36]. The CRP level is consistent over time. It is not affected by hematocrit or other blood-protein levels [46]. Moreover, it is unaffected by the circadian rhythm, time of food intake, blood sampling, anticoagulants, delay in specimen processing up to 6 h, or storage conditions. Nor is it affected by the specimen type: CRP measurements gave similar results in fresh, thawed, and even repeatedly thawed and refrozen plasma and serum. CRP can be measured using standard laboratory methods such as enzyme-linked immunosorbent assay (ELISA), turbidimetry, or nephelometry without significant discrepancies in results [46]. Thus, CRP is a convenient laboratory biomarker widely used in clinical practice and basic research.

CRP levels rise manifold in response to infection or tissue damage: from 5–10 mg/L in mild cases to 320–550 mg/L in the most severe cases [47,48]. However, in atherosclerosis CRP levels are usually below 5 mg/L. A high-sensitivity assay with a threshold of 0.28 mg/L was developed to measure CRP below this level [49]. Large prospective observational studies have demonstrated that in surveyed populations, CRP levels were within tertiles of less than 1.0 mg/L, 1–3 mg/L, and more than 3.0 mg/L. In meta-analyses, the odds ratio for MACE between the lower and upper tertiles was between 1.58 and 2.0 [50,51]. The PROVE IT-TIMI 22 trial demonstrated that, in patients on aggressive statin therapy, the median CRP level was 2.0 mg/L. Patients with a CRP level of 2.0 mg/L or more had a 30% higher relative risk of MACE [52]. Similar CRP-level medians and cardiovascular risk ratios were observed in subsequent large clinical trials of statin therapy [53,54]. Currently, a CRP level 2.0 mg/L or more is suggested by the American College of Cardiology/American Heart Association guidelines on cardiovascular disease prevention as a cardiovascular risk factor [9].

The association between the CRP level and MACE rate has been examined in large observational studies in postmenopausal women [55], healthy volunteers in the Physicians’ Health Study [56], and MRFIT [57]. A meta-analysis of 52 prospective studies that included 246,669 individuals without cardiovascular disease showed that increased CRP levels worsened the 10-year prognosis of cardiovascular risk [58]. In addition, a meta-analysis of the East Asian population showed an association between elevated CRP and higher cardiovascular risk [59]. Furthermore, the USPSTF meta-analysis that explored studies published from 1966 to 2007 demonstrated that relative cardiovascular risk is 1.58-fold higher in individuals with a CRP level more than 3.0 mg/L than in those with a CRP level less than 1.0 mg/L [50].

### 3.2. Pentameric C-Reactive Protein

pCRP belongs to the pentraxin family of acute-phase proteins. It is the primary acute-phase reactant in humans. pCRP is synthesized in hepatocytes and secreted into the bloodstream upon stimulation with IL-6. Circulating pCRP consists of five monomeric subunits bound with disulfide bonds in a ring-shaped disk [60]. Each subunit of the pentameric disk has a calcium-dependent binding site for lysophosphatidylcholine on one side and the complement component C1q on the other [61,62].

Phosphatidylcholine is a major structural component of cell and extracellular vesicle membranes. It is mainly present on the outer leaflet of the membrane phospholipid bilayer [63]. Secretory phospholipase A2 (PLA2) hydrolyzes it to lysophosphatidylcholine. Normally, phosphatidylcholine does not interact with PLA2. However, during apoptosis and cell injury, phospholipids of the inner leaflet translocate to the outer leaflet of the cell membrane. These phospholipids include phosphatidylserine and phosphatidylethanolamine, which are PLA2 ligands. In the presence of these two phospholipids, PLA2 hydrolyzes phosphatidylcholine to biologically active lysophosphatidylcholine [63]. In oxidized LDL, lipoprotein-associated PLA2 cleaves phosphatidylcholine in the lipid monolayer to lysophosphatidylcholine [64].

Lysophosphatidylcholine stimulates the endothelial synthesis of several chemokines, impairs endothelium-dependent arterial relaxation, increases oxidative stress, suppresses endotheliocyte migration and proliferation, and facilitates macrophage activation and polarization to the inflammatory M1 phenotype [65]. Lysophosphatidylcholine of oxidized LDL contributes to lysosomal damage and the NLRP-3 inflammasome activation in foam cells [66]. Lysophosphatidylcholine can activate the NLRP-3 inflammasome in adipose tissue, contributing to the development of insulin resistance [67].

Lysophosphatidylcholine is a ligand for pCRP [68]. Circulating pCRP acts as an opsonin that binds to lysophosphatidylcholine on the surface of cell membranes and oxidized lipoproteins. The sites on the reverse side of the CRP disc interact with the complement component C1q. This results in activation of the classical complement cascade up to the C4 component. Thus, CRP-induced complement activation facilitates phagocytosis of damaged cells and oxidized lipoproteins but does not initiate the formation of the membrane-attack complex C5b–C9 [68]. pCRP can also interact with factor H and activate the complement cascade up to the C4 component via the alternative pathway [69]. Furthermore, pCRP can opsonize nuclear antigens released by apoptotic and necrotic cells [70]. Therefore, the biological role of circulating pCRP is characterized by the facilitation of the clearance of cell-destruction products formed during trauma, infection, or sterile inflammation.

### 3.3. Monomeric C-Reactive Protein

Upon binding to lysophosphatidylcholine, the pentameric disk of CRP undergoes dissociation through intermediate forms into the final product, mCRP [71,72]. Dissociation occurs through the disintegration of disulfide bonds between the pCRP subunits [73]. This process involves lysophosphatidylcholine but also requires other cell-membrane components and calcium. Soluble lysophosphatidylcholine does not dissociate pCRP in the absence of cell membranes [72]. Dissociation opens a neoepitope (octapeptide Phe-Thr-Lys-Pro-Gly-Leu-Trp-Pro) on the C-terminal end of monomeric subunits, which is concealed in the pentameric disk [72]. This dramatically changes the antigenic specificity and biological functions of CRP [74].

mCRP has reduced aqueous solubility and remains predominantly bound to the cell membranes [75]. It has been detected in extracellular vesicles circulating in the bloodstream. A pronounced increase in the number of mCRP-positive extracellular vesicles has been observed in patients with acute myocardial infarction [76] and peripheral artery disease [77]. mCRP in monocyte-derived exosomes has also been detected in patients with stable coronary artery disease [78].

mCRP may contribute to thromboinflammation. Thromboinflammation has recently been introduced to describe the complex interplay between blood coagulation and inflammation [79]. Immobilized on a collagen substrate, mCRP substantially increased platelet adhesion and thrombus growth rate at the shear rate of 1500 s^−1^, characteristic of arteries with mild stenosis [80]. Perfusion of mCRP-preincubated whole blood over a collagen type I-coated flow chamber yielded a similar result [80]. Unlike pCRP, mCRP induced platelet glycoprotein (GP) IIb/IIIa activation in a dose-dependent manner. mCRP facilitated platelet adhesion via activation of GP IIb/IIIa receptors. Moreover, pCRP dissociated into mCRP on activated adhered platelets during the perfusion of whole blood through a flow chamber [81]. Without dissociation, pCRP did not stimulate platelet adhesion or thrombus growth [80,81]. pCRP was attached to platelet membranes and dissociated into mCRP in another experiment with perfusion of whole blood over a surface coated with activated adhered platelets. GP IIb/IIIa inhibition with an antibody (abciximab) prevented pCRP dissociation [82]. mCRP stimulated platelet adhesion to the endothelial cells [83] and induced tissue-factor expression and fibrin formation on endothelial cells [84].

When dissociated on platelets and adhering to the vessel wall, mCRP can induce endothelial activation and leukocyte recruitment. mCRP enhanced endothelial activation and neutrophil attachment to the endothelium [83,85]; monocyte adhesion to the collagen [86], fibrinogen [87], and fibronectin matrix [88]; and T-lymphocyte extravasation [89]. In vitro, mCRP decreased nitric-oxide release and increased production of proinflammatory IL-8 and monocyte chemoattractant protein-1 by endothelial cells via the NF-kB pathway [90]. Moreover, mCRP stimulated leukocyte recruitment to the vessel wall, inducing the expression of vascular cell adhesion molecule-1, intercellular adhesion molecule-1, and E-selectin, as well as the production of IL-6 and IL-8 by the endothelium [83,90,91]. mCRP induced IL-8 production [75,92] and prevented neutrophil apoptosis [75]. In addition, mCRP stimulated macrophage and T-cell polarization to inflammatory M1 and Th1 phenotypes [93]. mCRP stimulated oxidized LDL uptake by macrophages [94]. The in vivo evidence that mCRP can stimulate monocyte infiltration into damaged tissues was obtained from recent studies on a murine model of myocardial infarction [95] and a rat model of renal ischemia/reperfusion injury [96]. Compared with controls, the infarcted myocardium of mCRP-pretreated mice demonstrated increased accumulation of macrophages of the inflammatory M1 phenotype [95]. Furthermore, the damaged renal tissue of pCRP-pretreated rats demonstrated increased infiltration with monocytes colocalized with mCRP [96]. In addition, mCRP has been shown to stimulate neoangiogenesis and stabilize novel microvessels in vitro (bovine aortic endothelial cells and SMC) and in vivo (chorioallantoic membrane) [97,98]. In summary, the role of mCRP has been suggested in: platelet activation, adhesion, and aggregation; endothelial activation; leukocyte recruitment and polarization; foam-cell formation; and neovascularization (Figure 1).

The problem of mCRP deposition into atherosclerotic plaques has been addressed in several immunohistochemical studies. Herein, we discuss studies of human tissues. mCRP deposits have been detected in atherosclerotic plaques of the aorta [86], carotid [86,87,99], coronary [100,101], and femoral arteries [102], as well as diseased coronary artery venous bypass grafts [103]. Furthermore, mCRP deposits have been found in inflamed human striated muscles and infarcted myocardium [87]. CRP colocalized with leukocytes in thrombotic masses. CRP deposits in atheromatous tissues were larger in patients with increased CRP levels in blood plasma [100]. CRP accumulation in carotid atherosclerotic plaques was lower in patients treated with aspirin, angiotensin-converting enzyme inhibitors, or angiotensin-receptor blockers than in those who did not receive this therapy [104]. mCRP was deposited predominantly in the necrotic core and around clusters of macrophages, T-cells, and SMC, as well as neovessels in atherosclerotic plaques. Notably, CRP deposits were not found in intact arteries or fibrous or calcific plaques [86,87,99,100,102,103,104].

Some of the mentioned studies clearly distinguished between the two forms of CRP and confirmed that mCRP, but not pCRP, was deposited into damaged tissues [86,87,101], whereas other studies did not discriminate between CRP forms [99,100,102,103,104]. Nevertheless, it is unclear how mCRP accumulates in tissues. It can cross the endothelial barrier after dissociation [87] or be synthesized locally. The problem of local mCRP production has been explored in several studies. However, most of these studies did not discriminate between CRP forms. This creates difficulties in interpreting the locally synthesized form. An in vitro study demonstrated CRP production in an adipocyte culture stimulated by proinflammatory cytokines [105]. The expression of membrane-associated mCRP in human monocytes and CRP mRNA expression has been shown in another study [106]. Lipopolysaccharide-stimulated cultured macrophages synthesized mCRP and expressed CRP mRNA [107]. CRP mRNA expression in lipopolysaccharide-stimulated cultured macrophages was also observed in our study [78]. In an ex vivo study, monocytes collected from lipopolysaccharide-treated volunteers expressed CRP mRNA and produced CRP [108]. Moreover, CRP mRNA expression was detected in SMC and endotheliocytes from atheromatous tissues of coronary arteries [100], in atheromatous tissues of the femoral [102] and carotid [104] arteries, and diseased coronary artery venous bypass grafts [103]. In contrast, tissues of intact arteries did not express CRP mRNA [100,102,103,104]. Nonetheless, the contribution of local synthesis to the total concentration of mCRP in the tissues and bloodstream is unknown.

### 3.4. Level of Monomeric C-Reactive Protein in Health and Disease

To date, nine studies on the measurement of mCRP levels in human serum or plasma have been published (Table 1).

Wang et al. measured mCRP levels in 101 patients with acute myocardial infarction, 38 with unstable angina, 41 with stable angina, and 43 without coronary artery disease [109]. The mCRP level was 20.96 ± 1.64 μg/L in patients with myocardial infarction and 0.0 μg/L in other groups. No difference was found in mCRP levels between patients with ST-segment elevation and non-ST-segment-elevation myocardial infarction. mCRP level was substantially higher in patients deceased within 30 days from the onset of myocardial infarction than in survivors (36.70 ± 10.26 μg/L vs. 19.41 ± 1.43 μg/L) [109].

Zhang et al. measured mCRP levels in patients with autoimmune skin disorders (urticaria, eczema, and psoriasis; 20 patients in each group) and 20 healthy volunteers [110]. The mCRP level in patients with urticaria was 59.8 (44.5; 79.1) μg/L; psoriasis, 35.4 (17.0; 48.6) μg/L; and eczema, 30.0 (4.77; 36.3) μg/L, whereas in controls it was lower: 15.2 (5.05; 27.1) μg/L [110].

Williams et al. measured mCRP levels in 40 patients with markedly elevated CRP levels (more than 100 mg/L). The mean mCRP level in samples was 1.03 mg/L (±0.11 SE) [111].

Wu et al. measured mCRP levels in 37 patients with anti-neutrophil cytoplasmic antibody-associated vasculitis (AAV) and 20 control participants [112]. The mCRP level was 244.1 (226.1; 331.7) mg/L in patients with AAV and 170.0 (135.7; 199.3) mg/L in controls. The highest mCRP level was detected in four patients with AAV who had myocardial infarction: 581.4 (508.7; 647.3) mg/L and 240.8 (219.2; 292.1) mg/L in patients with and without myocardial infarction, respectively [112].

Munuswamy et al. measured mCRP levels in 38 patients with chronic obstructive pulmonary disease (COPD) and 18 non-COPD individuals. mCRP level was 0.66 (0.38; 1.03) mg/L in patients with COPD and 0.0 (0.0; 0.29) mg/L in non-COPD controls [113].

Liang et al. measured the mCRP levels in 206 patients with osteoarthritis and 60 healthy controls [114]. The mCRP level was 12.5 (7.8; 24.8) μg/L in patients with osteoarthritis and 5.0 (3.5; 9.8) μg/L in controls. Higher mCRP levels were associated with a more severe course of osteoarthritis [114].

Melnikov et al. reported a 7-year follow-up study of 80 patients with subclinical carotid atherosclerosis and an initially moderate cardiovascular risk [115]. The median mCRP level of the entire cohort was 5.2 (3.3; 7.1) μg/L. Patients with a median mCRP level or higher were more likely to have an increased number and total height of carotid atherosclerotic plaques at the end of the 7-year follow-up [115].

Karlsson et al. measured mCRP levels in 160 patients with systemic lupus erythematosus and 30 with AAV [116]. mCRP level was 3.7 (1.3; 7.4) μg/L in patients with systemic lupus erythematosus and 11 (5.8; 22) μg/L in those with AAV. No association was found between mCRP levels and any type of irreversible organ damage [116].

Fujita et al. measured the mCRP levels in 70 patients with autoimmune skin diseases, 50 with infectious diseases, and 30 healthy controls [117]. In the autoimmune skin-disease group, the mean mCRP levels were 477 (100; 2570) μg/L in patients with adult-onset Still’s disease, 186 (0.035; 626) μg/L in those with polymyalgia rheumatica, and 77 (0.035; 501) μg/L in those with rheumatoid arthritis. The mCRP level was 228 (0.035; 1086) μg/L in the infection group and 1.2 (0.035; 10) μg/L in the control group [117].

It is noteworthy that no correlation between mCRP and hsCRP level was found in five out of nine mentioned studies [111,113,115,116,117], whereas in the study by Wu et al., it was negative [112]. In contrast, Liang et al. reported a strong positive correlation between mCRP and hsCRP in patients with rheumatoid arthritis [114]. The studies by Zhang et al. and Wang et al. did not specifically examine this correlation [109,110].

None of these studies explored the correlation between mCRP levels and other proteins of the pentraxin family, in particular, pentraxin 3, which is considered a biomarker of cardiovascular disease [118]. It is synthesized locally by a range of stromal and myeloid cells in response to tissue damage and inflammation. It was suggested that pentraxin 3 and hsCRP could be measured complementary to assess local and systemic inflammatory response. Pentraxin 3 and hsCRP levels rarely correlate [118]. The association between pentraxin 3 and mCRP requires further research.

In summary, these studies demonstrated the feasibility of measuring mCRP levels. Most of these studies reported mCRP values in a comparable range. The nonconforming data on the correlation between mCRP and hsCRP levels may be at least partially explained by the different conditions in which they were analyzed. Some studies have reported an association between mCRP levels and disease severity and outcome. Nevertheless, large prospective studies are required to provide sufficient evidence to support this association or even establish mCRP levels as a diagnostic or prognostic tool.

## 4. Treatment to Reduce CRP Level and Ameliorate Residual Inflammatory Risk

### 4.1. Lifestyle Modifications

Lifestyle interventions can decrease subclinical vascular inflammation. According to the 10-year UCC-SMART study, smoking cessation, body-weight reduction, and regular physical activity decreased CRP levels [119]. Regular aerobic exercise decreased CRP levels independently of body-weight reduction or statin therapy [120]. Moreover, a meta-analysis of 83 studies including 3769 patients demonstrated the effect of regular physical activity on CRP levels [121]. A meta-analysis of 76 studies involving 6742 patients with obesity confirmed the association between body-weight reduction and a decrease in proinflammatory cytokines and CRP levels [122]. Furthermore, the consumption of foods rich in wholegrain cereals, dietary fiber, omega-3 fatty acids, and vitamins E and C reduced CRP levels [120].

### 4.2. Statins

Randomized clinical trials have demonstrated that statins influence subclinical vascular inflammation. The PROVE IT-TIMI 22 and IMPROVE-IT trials showed that a CRP-level reduction to less than 2.0 mg/L decreased MACE rate by 28–33%, which was virtually identical to patients with a LDL-C-level reduction of less than 1.8 mmol/L [52,123]. In the REVERSAL trial, the slowest progression of atheroma occurred in patients with LDL-C- and CRP-level reductions above the median values [124]. The JUPITER trial demonstrated the efficacy of statin therapy in the primary prevention of atherosclerosis in individuals with an initial CRP level of 2.0 mg/L or more and an LDL-C level less than 3.4 mmol/L, resulting in a 44% reduction in the relative risk of MACE [125]. In the FOURIER trial, the MACE rate was associated with a CRP level independent of LDL-C. In the group that achieved an LDL-C level of less than 0.52 mmol/L on high-dose statin plus evolocumab, MACE occurred in 9.0% of patients with a CRP level less than 1.0 mg/L vs. 13.1% of patients with a CRP level more than 3 mg/L [126]. The rate of MACE events per 100 human-years in the SPIRE-1/SPIRE-2 trials was 1.96 in patients with a CRP level less than 1.0 mg/L vs. 3.59 in patients with a CRP level more than 3.0 mg/L [127]. Two retrospective studies of patients undergoing percutaneous coronary intervention also demonstrated that MACE occurred less frequently in patients with initially low CRP levels or those who responded to statin therapy with a reduction in CRP level below 2.0 mg/L [128,129].

Although statins ameliorate subclinical vascular inflammation, their effect is sufficient only in approximately half of intensively treated patients. Despite aggressive lipid-lowering with statins, CRP levels remained above 2.0 mg/L in 43% of patients in PROVE IT-TIMI 22, 47% in IMPROVE-IT, 47% in SPIRE-1/SPIRE-2 [130], 43.6% in ODYSSEY OUTCOMES [131], and 45.6% and 48% in percutaneous coronary intervention trials [128,129]. The prevalence of residual inflammatory risk is higher in the general population [132].

### 4.3. Other Lipid-Lowering Agents

The IMPROVE-IT trial showed that the combination of statins with ezetimibe more often led to the achievement of both LDL-C and CRP target levels, which was associated with the lowest MACE rate. However, this effect was ascribed to the achievement of combined target levels, not direct ezetimibe action [123]. Ezetimibe monotherapy had no effect on CRP levels [133,134,135]. The addition of ezetimibe to statin therapy did not result in an additional reduction in CRP levels [136]. PCSK9 blockers also do not affect inflammatory activity or alter CRP levels [137,138].

### 4.4. Anti-Inflammatory Agents

#### 4.4.1. Upstream Agents

The data from randomized clinical trials of anti-inflammatory agents support the hypothesis that MACE reduction can be achieved by directly targeting the NLRP3 inflammasome and IL-1β/IL-6/CRP axis (Figure 2). In contrast, agents that did not affect this inflammatory cascade failed to alter the MACE rate.

The CIRT trial studied the efficacy of low-dose methotrexate in the secondary prevention of atherosclerosis in patients with type 2 diabetes mellitus or metabolic syndrome [139]. The initial CRP level in the patients was 1.5 mg/L. During the 2.3-year follow-up, methotrexate neither reduced IL-1β, IL-6, or CRP levels nor altered the MACE rate. A possible explanation for the ineffectiveness of methotrexate is that a low baseline CRP level indicated an initially low residual inflammatory risk in patients. Moreover, the mechanism of the anti-inflammatory action of methotrexate does not affect the NLRP3 inflammasome and IL-1β/IL-6/CRP axis [36]. Two other anti-inflammatory agents that do not affect the NLRP3 inflammasome and the IL-1β/IL-6/CRP axis also failed to reduce MACE in patients with acute myocardial infarction [36]. These agents are losmapimod, a p38 MAPK inflammatory pathway inhibitor studied in the LATITUDE-TIMI 60 trial [140] and darapladib, a lipoprotein-associated PLA2 inhibitor studied in the SOLID-TIMI 52 trial [141].

Conversely, colchicine, which inhibits the NLRP3 inflammasome, was effective in reducing MACE. This anti-inflammatory agent is routinely used to suppress crystal-mediated inflammation in gout and Mediterranean fever [142]. The COLCOT trial showed that low-dose colchicine treatment resulted in a 23% reduction in the relative risk of MACE in patients with acute myocardial infarction [143]. Furthermore, the LoDoCo2 trial demonstrated that low-dose colchicine treatment reduced the relative risk of MACE by 31% in patients with stable coronary artery disease [144]. Nevertheless, the COPS trial failed to demonstrate a reduction in MACE in patients with acute myocardial infarction following low-dose colchicine treatment [145]. Colchicine ineffectiveness in the COPS trial could be related to the clinical characteristics of the study group, drug dosage, and time of therapy initiation [145].

The CANTOS trial examined the efficacy of the IL-1β antagonist canakinumab for the secondary prevention of atherosclerosis in patients with CRP levels of 2 mg/L or more [146]. Canakinumab reduced CRP and IL-6 levels by 36–40% without altering the lipid profile. The canakinumab group demonstrated a 14.2% reduction in the relative risk of MACE [146]. The subgroup analysis showed a 25% MACE reduction in patients who achieved a CRP level of less than 2.0 mg/L and no MACE reduction in non-responders to canakinumab with a CRP level of 2.0 mg/L or more compared with that in the placebo group [147]. A decrease in IL-6 level below the median of 1.65 ng/L on canakinumab treatment was associated with a 32% reduction in the relative risk of MACE [148].

The RESCUE trial studied the efficacy and safety of ziltivekimab, a monoclonal antibody against IL-6, in patients with an hsCRP level of 2.0 mg/L or more and moderate to severe chronic kidney disease [149]. Twelve weeks of treatment resulted in a dose-dependent 77–92% reduction in the median CRP level. Ziltivekimab also produced a dose-dependent reduction in the levels of fibrinogen, serum amyloid A, haptoglobin, secretory phospholipase A2, and lipoprotein(a) [149]. In late 2021, the ZEUS trial was initiated to test whether ziltivekimab treatment of patients with cardiovascular disease, chronic kidney disease, and elevated CRP levels would result in a reduction in the MACE rate [150].

#### 4.4.2. Anti-CRP Agents

Attempts to ameliorate inflammation by directly influencing CRP (Table 2) aim at one of three targets: translation of CRP, dissociation of pCRP into mCRP, and mCRP.

One approach is to use antisense oligonucleotides that promote degradation of CRP mRNA and prevent CRP translation. Treatment with an antisense oligonucleotide ISIS 280290 reduced CRP levels in a rabbit model of atherosclerosis but did not result in an antiatherogenic effect [151]. In another study, treatment with an antisense oligonucleotide ISIS 329993 produced a pronounced decrease in CRP levels and ameliorated collagen-induced arthritis symptoms in CRP transgenic mice. In addition, it substantially reduced CRP levels and was well tolerated in healthy volunteers with initial CRP levels between 2 and 10 mg/L [152]. Additionally, ISIS 329993 administration markedly reduced CRP levels in endotoxin-pretreated healthy volunteers [153]. Nevertheless, it did not reduce atrial fibrillation burden in patients with paroxysmal atrial fibrillation in a phase II trial [154]. Furthermore, its effectiveness was unclear in a phase II trial of patients with rheumatoid arthritis [155].

Another approach is to target pCRP-mCRP dissociation. 1,6-bis (phosphocholine)-hexane (1,6-bisPC) inhibits pCRP to mCRP dissociation by cross-linking two pCRP disks in a “decameric” form that conceals phosphatidylcholine-binding sites. In a rat model of ischemia/reperfusion injury, this agent abrogated mCRP deposition and reduced leukocyte infiltration into the damaged myocardium [87]. In another study, 1,6-bisPC administration decreased infarct size and cardiac dysfunction induced by the injection of human CRP into rats with induced acute myocardial infarction [156]. Furthermore, 1,6-bisPC administration decreased the inflammatory response in damaged renal tissue and improved blood-urea nitrogen levels in a rat model of renal ischemia/reperfusion injury [96]. However, the suboptimal pharmacokinetic profile of 1,6-bisPC hinders its further development as an anti-inflammatory agent [87,156]. A recent study introduced a novel phosphatidylcholine-mimetic compound, C10M, that binds to phosphatidylcholine-specific sites on the pCRP disk [157]. This prevents pCRP from interacting with phosphatidylcholine and disrupts dissociation. In vitro, the C10M compound decreased pCRP binding to activated platelets, mCRP-induced endotheliocyte activation, and monocyte adhesion. In a rat model of CRP-induced aggravation of renal ischemia/reperfusion injury, C10M abrogated mCRP deposition in the damaged tissues and improved excretory renal function. Furthermore, this novel compound inhibited CRP-mediated allograft rejection in a rat model of hindlimb transplantation [157]. The current form of this compound may need further development, considering its relatively low bioavailability and short half-life [158].

The third approach is to directly target mCRP. A monoclonal antibody against mCRP attenuated rheumatoid arthritis symptoms, joint inflammation, pannus formation, and bone destruction in a murine arthritis model [159]. The same antibody also ameliorated glomerular damage and the progression of proteinuria in a murine lupus nephritis model [159].

To date, anti-inflammatory therapy has targeted the NLRP3 inflammasome and upstream links of the IL-1β/IL-6/CRP axis. Large clinical trials have provided sufficient evidence to support this strategy. Few known compounds target CRP. Studies on these agents are limited to basic research and animal models. Antisense oligonucleotides that target CRP mRNA were studied in small clinical trials; however, they did not demonstrate a clinical benefit. It is not clear whether targeting CRP might provide an advantage over upstream agents. On the one hand, upstream targeting reduces CRP level as well, and this is explicitly demonstrated in clinical trials. In contrast, mCRP production at sites of local inflammation (either by local synthesis or dissociation of circulating pCRP) may be independent of the overall CRP production in the liver. In this case, targeting mCRP or pCRP-mCRP dissociation might prove effective. However, to date, no trials have explored this strategy.

## 5. Conclusions

pCRP is the main biomarker of inflammation that reflects the activity of the IL-1β/IL-6/CRP axis. mCRP is a form of CRP with a distinct proinflammatory profile. mCRP has long remained the subject of basic research; however, recent studies have introduced mCRP assays and demonstrated the feasibility of mCRP-level measurement. Notably, most studies have reported mCRP values in a comparable range. Some studies have reported an association between mCRP levels and disease severity and outcome. Nevertheless, large prospective studies are required to provide sufficient evidence to support this association or even establish mCRP levels as a diagnostic or prognostic tool. Regarding anti-inflammatory therapy, there are few known compounds that target CRP. Studies on these agents are limited to animal models or small clinical trials.

## Figures and Tables

**Figure 1 ijms-24-02079-f001:**
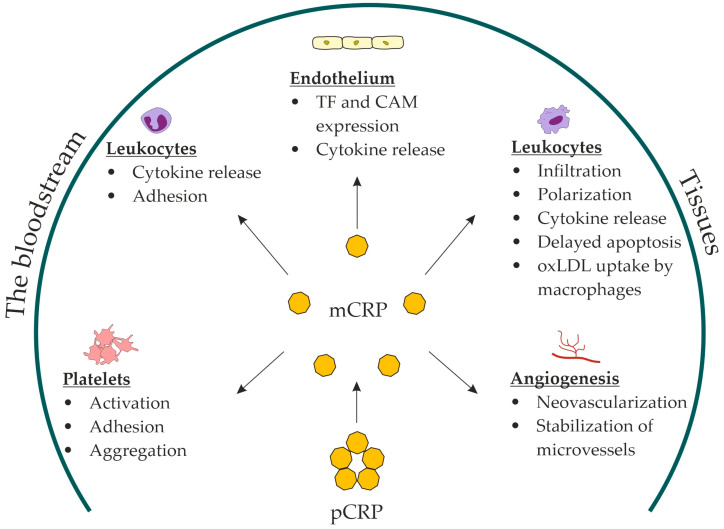
Proposed roles for mCRP in atherosclerosis. mCRP, monomeric C-reactive protein; pCRP, pentameric C-reactive protein; TF, tissue factor; CAM, cell-adhesion molecules; oxLDL, oxidized low-density lipoproteins.

**Figure 2 ijms-24-02079-f002:**
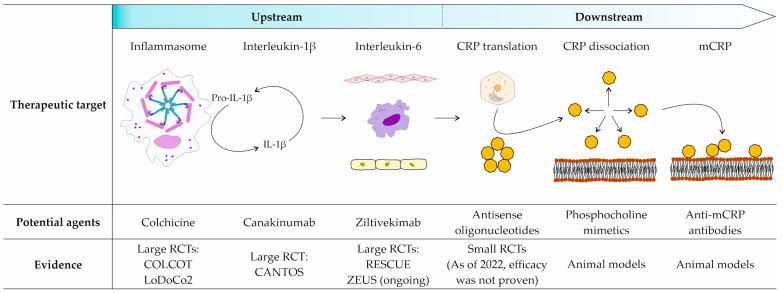
Anti-inflammatory agents targeting the NLRP3 inflammasome and IL-1β/IL-6/CRP axis. Upstream agents targeting the NLRP3 inflammasome, interleukin (IL)-1β, and IL-6 reduce the rate of major adverse cardiovascular events, according to large RCTs. Downstream agents target translation of pCRP in the liver, CRP dissociation, and mCRP. Studies on these agents are limited to animal models and small RCTs. CRP, C-reactive protein; mCRP, monomeric C-reactive protein; pCRP, pentameric C-reactive protein; RCTs, randomized clinical trials.

**Table 1 ijms-24-02079-t001:** Monomeric C-reactive protein assays.

Assay Type	Anti-mCRP Antibody	Blood-Specimen Type	Reported mCRP Levels	Correlation (mCRP/ hsCRP)	Reference
ELISA	Unspecified mAb	Heparinized plasma	20.96 ± 1.64 μg/L in acute myocardial infarction 0.0 μg/L in angina pectoris, unstable angina, and non-CAD patients	n/a	Wang et al., 2015 [109]
ELISA	CRP-8	Plasma (unspecified anticoagulant)	59.8 (44.5; 79.1) μg/L in urticaria 35.4 (17.0; 48.6) μg/L in psoriasis 30.0 (4.77; 36.3) in eczema 15.2 (5.05; 27.1) μg/L in healthy participants	n/a	Zhang et al., 2018 [110]
ELISA	CRP-8	Serum purified by chromatography	1.03 ± 0.11 mg/L in patients with hsCRP > 100 mg/L	None	Williams et al., 2020 [111]
ELISA	Unspecified mAb	EDTA plasma	244.1 (226.1; 331.7) mg/L in AAV 170.0 (135.7; 199.3) mg/L in control participants	Negative	Wu et al., 2020 [112]
ELISA	Aptamer	Serum	0.66 (0.38; 1.03) mg/L in COPD 0.0 (0.0; 0.29) mg/L in non-COPD patients	None	Munuswamy et al., 2021 [113]
ELISA	CRP-8	Plasma (unspecified anticoagulant)	12.5 (7.8; 24.8) μg/L in osteoarthritis 5.0 (3.5; 9.8) μg/L in healthy participants	Positive	Liang et al., 2022 [114]
CBA	CRP-8	Citrated plasma	5.2 (3.3; 7.1) μg/L in patients with subclinical carotid atherosclerosis	None	Melnikov et al., 2022 [115]
ELISA	8C10	Serum	3.7 (1.3; 7.4) μg/L in systemic lupus erythematosus 11 (5.8; 22) μg/L in AAV patients	None	Karlsson et al., 2022 [116]
ELISA	12C	EDTA plasma	477 (100; 2570) μg/L in adult-onset Still’s disease 186 (0.035; 626) μg/L in polymyalgia rheumatica 77 (0.035; 501) μg/L in rheumatoid arthritis 228 (0.035; 1086) μg/L in acute infections 1.2(0.035; 10) μg/L in healthy participants	None	Fujita et al., 2022 [117]

ELISA, enzyme-linked immunosorbent assay; CBA, cytometric bead array; mAb, monoclonal antibody; mCRP, monomeric C-reactive protein; hsCRP, pentameric C-reactive protein measured using high-sensitivity assay; EDTA, ethylenediaminetetraacetic acid; CAD, coronary artery disease; AAV, anti-neutrophil cytoplasmic antibody-associated vasculitis; COPD, chronic obstructive pulmonary disease; n/a, not assessed.

**Table 2 ijms-24-02079-t002:** CRP-targeting agents.

Target	Agent	Stage of Development	Main Findings
CRP translation	ISIS 280290	Animal models	No antiatherogenic effect was shown in a rabbit model of atherosclerosis.
	ISIS 329993	Phase II clinical trials	No benefit was shown in atrial fibrillation or rheumatoid arthritis treatment.
CRP dissociation	1,6-bisPC	Animal models	Myocardial damage and renal damage were reduced in rat models of myocardial and renal IRI, respectively.
	C10M	Animal models	Renal damage was reduced and allograft rejection prevented in rat models of renal IRI and hindlimb transplantation, respectively.
mCRP	mAb C3	Animal models	Joint damage and renal damage were reduced in murine models of rheumatoid arthritis and lupus nephritis, respectively.

CRP, C-reactive protein; mCRP, monomeric C-reactive protein; 1,6-bisPC, 1,6-bis(phosphocholine)-hexane; mAb, monoclonal antibody; IRI, ischemia/reperfusion injury.

## Data Availability

Not applicable.

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
