# Peer review of "Monomeric C-Reactive Protein in Atherosclerotic Cardiovascular Disease: Advances and Perspectives"

_ijms, 2023, doi:10.3390/ijms24032079_

Round 1

Reviewer 1 Report

This is a comprehensive review describing the implication of the monomeric C-reactive protein (mCRP) in the inflammatory pathway NLRP3 inflammasome in the atherosclerotic cardiovascular diseases.

The authors presented a well review of all the forms of CRP measurement in vitro and in vitro. This important with regards to potential bias of some of studies outcome related to specific technique and CRP form measured. The authors also highlighted the importance of mCRP determination in patients with CVD and other various inflammatory diseases. Also, the authors discussed the potential of anti-inflammatory therapy as well in atherosclerosis that targeted the NLRP3 inflammasome and upstream links of the IL-1β/IL-6/CRP axis. Overall, the review is well written, constructive and very informative.

Author Response

We are grateful to you for reviewing this manuscript.

Reviewer 2 Report

The authors present a review of the current status of the research of monomeric C-reactive protein in atherosclerosis. The review has a good scientific sounding and clear structure.

The contemporary concepts of the mechanisms of vascular inflammation are discussed with the engagement of the most significant sources in recent years. The results of clinical studies on the prognostic significance of inflammatory markers in atherosclerotic diseases, as well as studies of promising anti-inflammatory agents, are discussed. The fundamental aspects of biology and immunology of monomeric C-reactive protein, the potential of its use for diagnostic and prognostic purposes, as well as as a therapeutic target are described in detail.

I have one question, the coverage of which in the article in my opinion would be important. The authors discuss the potential advantages of measuring mCRP in comparison with pCRP. Perhaps other members of the pentraxin family, for which methods of determination have already been standardized, overlap the potential advantages of monomeric CRP? For example, pentraxin 3 can also be synthesized locally in the foci of inflammation and represent vascular inflammation in atherosclerosis. It also plays a direct role in the regulation of inflammation. Is there evidence of a relationship between mCRP and pentraxin 3 or other pentraxins, not only pCRP?

Author Response

Dear reviewer, thank you for your valuable commentary.

Point 1: I have one question, the coverage of which in the article in my opinion would be important. The authors discuss the potential advantages of measuring mCRP in comparison with pCRP. Perhaps other members of the pentraxin family, for which methods of determination have already been standardized, overlap the potential advantages of monomeric CRP? For example, pentraxin 3 can also be synthesized locally in the foci of inflammation and represent vascular inflammation in atherosclerosis. It also plays a direct role in the regulation of inflammation. Is there evidence of a relationship between mCRP and pentraxin 3 or other pentraxins, not only pCRP?

Response 1: we have added the following:

Lines 357-363: “None of these studies explored correlation between mCRP levels and other proteins of the pentraxin family, in particular, pentraxin 3. Pentraxin 3 is considered a biomarker of cardiovascular disease. It is synthesized locally by a range of stromal and myeloid cells in response to tissue damage and inflammation. It was suggested that pentraxin 3 and hsCRP could be measured complementary to assess local and systemic inflammatory response. Pentraxin 3 and hsCRP levels rarely correlate [Ristagno et al., 2019. DOI: 10.3389/fimmu.2019.00823]. Whether there is any association between pentraxin 3 and mCRP is a matter of future research”.

Reviewer 3 Report

In the present paper, Melnikov et al. conducted a comprehensive review evaluating the feasibility of monomeric CRP (mCRP) level measurement as a diagnostic or prognostic tool in the clinics. It also describes the pathway from NLRP3 inflammasome and inflammatory IL-1β/IL-6/CRP axis as well as the new strategies for anti-inflammatory therapy. Since pro-inflammatory mCRP has emerged as a critical player in atherosclerotic cardiovascular disease, this well-written review is of particular interest. I only have the following specific comments:

-          The fact that mCRP is able to bind to cell membranes and extracellular vesicles could affect the response to pharmacological treatments such as anti-platelet therapy?  Please argue this relevant issue.

-          Please expand the effects of mCRP on age-related diseases.

-          Given the role of mCRP on thromboinflammation, authors could speculate about the its potential involvement in coronavirus disease 19.

-          A figure summarising section 4 concerning anti-inflammatory therapeutic approaches is lacking.

-          What is missing for the development of technological advances in order to accurately measure mCRP in the serum or plasma and, therefore, mCRP measurement becoming clinically used?

Minor comments

-          Line 216: the ‘microparticle’ term is no longer used. ‘Extracellular vesicles’ should be used instead.

Author Response

Dear reviewer, thank you for your valuable remarks and suggestions.

Point 1: The fact that mCRP is able to bind to cell membranes and extracellular vesicles could affect the response to pharmacological treatments such as anti-platelet therapy?  Please argue this relevant issue.

Response 1:

We have mentioned the following:

Line 234: "GP IIb/IIIa inhibition with an antibody (Abciximab) prevented pCRP dissociation [De la Torre et al., 2013. DOI: 10.1111/jth.12415]"

Lines 272-275: “CRP accumulation in carotid atherosclerotic plaques was lower in patients treated with aspirin, angiotensin-converting enzyme inhibitors, or angiotensin receptor blockers than in those who did not receive this therapy [Sattler et al., 2005. DOI: 10.1161/01.STR.0000150643.08420.78]”.

There are no other studies known to us that explored interactions of mCRP and anti-platelet therapy.

Point 2: Please expand the effects of mCRP on age-related diseases.

Response 2: This relevant problem requires detailed discussion, which we believe is beyond the scope of this manuscript. Moreover, there are excellent reviews on mCRP role in a range of diseases, including age-related diseases. We introduced the references to these reviews in the ‘Introduction’ as follows:

Lines 67-69: “The role of mCRP in other conditions, including cancer, venous thromboembolism, age-related and neurodegenerative diseases, has been discussed elsewhere [Dix et al., 2022. DOI: 10.3389/fimmu.2022.1002652. Potempa et al., 2021. DOI: 10.3389/fimmu.2021.744129. McFadyen et al., 2020. DOI: 10.1007/978-3-030-41769-7_20. Luan et al., 2018. DOI: 10.3389/fimmu.2018.01302]”.

Point 3: Given the role of mCRP on thromboinflammation, authors could speculate about the its potential involvement in coronavirus disease 19.

Response 3: This is, indeed, a fascinating subject. Yet, our intent was to write a manuscript with a focus on mCRP role in atherosclerosis from pathophysiology to treatment. As atherosclerosis and COVID-19 are essentially different, we would have to substantially enlarge the manuscript and shift the focus from atherosclerosis. We believe that the detailed discussion of the role of mCRP in COVID-19 deserves a separate review.

Point 4: A figure summarising section 4 concerning anti-inflammatory therapeutic approaches is lacking.

Response 4: The figure was introduced (lines 421-428).

Point 5: What is missing for the development of technological advances in order to accurately measure mCRP in the serum or plasma and, therefore, mCRP measurement becoming clinically used?

Response 5: There is a range of problems. Assays may lack reproducibility. There is no single way for standardization and calibration, as different laboratories use different techniques for mCRP production (usually, urea treatment of pCRP in the absence of calcium, or production of recombinant mCRP). It is not clear if mCRP measurements in different blood specimen types (serum, EDTA/citrated/heparinized plasma) produce similar results. Neither is it studied how storage conditions and sample processing affects mCRP. These are at least some of the obstacles. We believe that these difficulties can be overcome in the near future, and mCRP assays will be introduced into clinical practice.

Point 6: Line 216: the ‘microparticle’ term is no longer used. ‘Extracellular vesicles’ should be used instead.

Response 6: The term ‘microparticles’ was substituted with ‘extracellular vesicles’ (lines 177, 217, 218).